# Wolf Dispersal Patterns in the Italian Alps and Implications for Wildlife Diseases Spreading

**DOI:** 10.3390/ani12101260

**Published:** 2022-05-13

**Authors:** Francesca Marucco, Kristine L. Pilgrim, Elisa Avanzinelli, Michael K. Schwartz, Luca Rossi

**Affiliations:** 1Department of Life Sciences and Systems Biology, University of Turin, Via Accademia Albertina 13, 10123 Turin, Italy; 2National Genomics Center for Wildlife and Fish Conservation, Rocky Mountain Research Station, USDA Forest Service, 800 E. Beckwith, Missoula, MT 59802, USA; kristine.pilgrim@usda.gov (K.L.P.); michael.k.schwartz@usda.gov (M.K.S.); 3Large Carnivore Center, Ente di Gestione Aree Protette Alpi Marittime, Piazza Regina Elena 30, Valdieri, 12010 Cuneo, Italy; elisa.avanzinelli@centrograndicarnivori.it; 4Department of Veterinary Sciences, University of Turin, L.go Braccini 2, 10095 Grugliasco, Italy; luca.rossi@unito.it

**Keywords:** dispersal, wolves, wildlife diseases, non-invasive genetic monitoring

## Abstract

**Simple Summary:**

Wildlife dispersal directly influences population expansion patterns, and may have indirect effects on the spread of wildlife diseases. For many species, little is known about dispersal, despite its importance to conservation. We documented the natural dispersal processes of an expanding wolf (*Canis lupus*) population in the Italian Alps to understand the dynamics of the recolonization pattern and identify diseases that might be connected with the process through the use of non-invasive genetic sampling over a 20-year period. By documenting 55 dispersal events, with an average minimum straight dispersal distance of 65.8 km (±67.7 km), from 7.7 km to 517.2 km, we discussed the potential implications for maintaining genetic diversity of the population and for wildlife diseases spreading.

**Abstract:**

Wildlife dispersal directly influences population expansion patterns, and may have indirect effects on the spread of wildlife diseases. Despite its importance to conservation, little is known about dispersal for several species. Dispersal processes in expanding wolf (*Canis lupus*) populations in Europe is not well documented. Documenting the natural dispersal pattern of the expanding wolf population in the Alps might help understanding the overall population dynamics and identifying diseases that might be connected with the process. We documented 55 natural dispersal events of the expanding Italian wolf alpine population over a 20-year period through the use of non-invasive genetic sampling. We examined a 16-locus microsatellite DNA dataset of 2857 wolf samples mainly collected in the Western Alps. From this, we identified 915 individuals, recaptured 387 (42.3%) of individuals, documenting 55 dispersal events. On average, the minimum straight dispersal distance was 65.8 km (±67.7 km), from 7.7 km to 517.2 km. We discussed the potential implications for maintaining genetic diversity of the population and for wildlife diseases spreading.

## 1. Introduction

Dispersal is a key component of the dynamics of spatially structured populations and in the expansion of species distributions, which drives recolonization patterns and the genetic structure of animal populations [1,2]. This, in turn, influences population viability [3]. In the last decades, large carnivore populations in Europe have been increasing and expanding [4], reinhabiting their former geographic range, however studies of dispersal patterns based on marked animals are still limited. A good understanding of species dispersal is important to predict population dynamics and to guide decision making for management and conservation.

Wolves began naturally recolonizing the southwestern Italian Alps at the beginning of the 1990s from the north Apennines wolf subpopulation [5], after being extirpated from the Alps in the early 1900s. Simulations of the wolf natural recolonization process showed that a total of 8–16 effective founders explained the genetic diversity observed in the western Alps in the first years of recolonization [5]. After 20 years, wolves reached higher densities in the western part of the Alps, expanding towards the central Alps [6]. Recently, a similar process of recolonization began in the eastern Alps as individuals from the Dinaric-Balkan population dispersed and reached the Alps [7]. This expansion is demonstrated by the case of a GPS collared male wolf from a Slovenian pack that traveled through Austria to finally settle with a female from Western Italy in the Italian Eastern Alps [8].

Wolf dispersal has been studied in several North American populations by using radio collars [9,10,11]. Dispersal patterns in Europe have also been demonstrated by using radiotelemetry or GPS technology [12,13,14,15,16]. In fewer studies, molecular genetic tools have been used for tracking the social dynamics of wolves and documenting dispersal events [17,18,19]. Non-invasive tools are getting widely used because they do not require physical captures of animals, and they allow the proper sampling of entire populations, even if distributed over a large scale. Dispersal rates are influenced by social factors such as population density, and also by habitat factors such as environmental characteristics and resource availability [2,20]. In wolves, offspring often disperse away from their natal pack to find a new breeding territory [21]. Dispersing wolves may travel over short or long distances from their natal area to find optimal habitat and establish their own pack [11]. Studies provide evidence that habitat barriers, environmental configuration, prey abundance, and individual characteristics influence dispersal pattern as well [20,22].

We examined the direction, distance of dispersal, and genetic patterns of expansion in the wolf recolonizing population in the Italian Alps over the last 20 years, using molecular genetics tools to analyze thousands of wolf samples collected in the Alps as part of a large and long-term study on wolf conservation in Italy [6]. Our primary objectives were to: (1) identify dispersers defined as genotypes who changed pack and territory, moving from an area to another one, documented by non-invasive genetic recaptures, (2) quantify straight-line dispersal distances and assess differences between sexes, frequency of status, and patterns of directions; (3) evaluate patterns of genetic variation; and (4) discuss how dispersal patterns might influence wildlife disease spreading and consider the management implications in this context.

## 2. Materials and Methods

Wolf samples were collected yearly between 2001 to 2021 across the Italian Alps over the wolf-occupied range as part of the Piemonte Region monitoring Program [23,24]. Genetic analysis on biological samples, mainly scats, but occasional dead wolf carcasses, or saliva swabs from wounds associated with predation events have been regularly conducted. A total of 5729 samples, including 95% feces, 0.31% hair samples, 4.10% tissue samples from dead animals, and 0.9% saliva swabs, were genotyped to identify individuals over the Alps between 2001 and 2021. Occasionally we received samples to analyze from neighboring countries (Switzerland, France, Slovenia, Germany), to test if the sampled genotype was previously detected in Italy, to document long distance dispersals, and to evaluate the origin of the individuals.

DNA was extracted from the various sample types primarily using kits and protocols from Qiagen (Valencia, CA, USA). Scat, hair, and saliva swabs were processed in a dedicated laboratory for non-invasive samples. Scats were processed using the QIAmp Fast DNA Stool Kit and saliva swabs using the QIAamp DNA investigator kit. Hair was processed using the DNeasy Blood and Tissue Kit (Qiagen, Valencia, CA, USA) adding 20 μL 1 M DTT and incubation modifications for hair [25]. DNA from tissues were extracted with the DNeasy Blood and Tissue Kit using the standard tissue protocol with overnight incubation. We amplified 697 bp of the left domain of the control region of mitochondrial DNA (mtDNA) for species and haplotype testing using primers L15926 and H576 [26,27]. Reaction volumes of 30 μL contained 50–100 ng DNA, 1 μM each primer, 1 U Amplitaq Gold DNA polymerase along with 1× PCR Buffer II and 2.5 mM MgCl2 (Life Technologies, New York, NY, USA), 200 μM each dNTP (New England Biolabs, Ipswich, MA, USA). We used an annealing temperature of 55 °C. PCR amplicons were run on a 1.6% agarose gel and only samples with PCR products that were visualized on the gel and in the correct size range were further purified using ExoSap-IT (Affymetrix-USB Corporation, Cleveland, OH, USA). The purified PCR products were sequenced bidirectionally using the PCR primers and internal primer L16007 [26] for non-invasive samples at Eurofins Genomics (Louisville, KY, USA). We used the program Sequencher (Gene Codes Corp., Ann Arbor, MI, USA) for sequence alignment, and the program Dambe [28] for haplotype determination.

We amplified 16 variable microsatellite loci, ten of which were used previously on wolves from this region [23]. We used the following loci: CPH2, CPH4, CPH5, CPH8, CPH12, FH2004, FH2054, FH2079, FH2088, FH2096, FH2137, FH2140, FH2161, CO9.250, C20.253, and Pez17 [29,30,31,32]. We amplified primers in five multiplex reactions each in a volume of (10 μL). We used 1.0–2.0 μL DNA template with 1 μM reverse primer, 1 μM dye-labeled forward primer, 1.5 mg/mL BSA, 200 μM each dNTP, 1 μL Amplitaq Gold DNA polymerase, 1× PCR Buffer II, and 2.0 mM MgCl2. DNA from non-invasive scat and hair samples was amplified at least twice at each locus using a multi-tube approach [33]. Microsatellite amplification products were visualized using a LI-COR DNA analyzer (LI-COR Biotechnology, Lincoln, NE, USA). Non-invasive samples that amplify for microsatellites were further tested for sex [34]. This sexing test targets the ZFX/ZFY region and yields two bands for males and one for females.

Microsatellite data was error checked for quality and potential allelic dropout and false alleles. We used allele scores only if they were consistent between amplifications, and samples were re-amplified at least twice more at loci with discrepancies until alleles were confirmed or were dropped from further analysis. Samples that failed at eight or more loci were removed as poor quality. We used the program Dropout 2.3 [35] to determine matching samples and calculate probability of identity. Unique genotypes were further tested for heterozygosity and allelic diversity GenAlEx [36]. We evaluated pack structure for paternal and maternal relationships using exclusion conducted by hand and subsequently using the program CERVUS 3.0 [37] using the strict (95%) confidence criteria to assess those relationships. Genotypes were also evaluated with ML-RELATE [38], where we tested individual relatedness to other individuals in the putative packs.

We defined dispersal as an event where a wolf genotype was captured in one location and subsequently recaptured in a different territory or new pack. In both cases the pack structure was determined with pedigree analysis. We monitored packs’ territories by means of snow tracking, photo trapping, and collection of presence signs, often verified by genetic analysis. Dispersal events were spatially analyzed using QGIS 3.24. [39]. The dispersal distance was estimated as the straight-linear distance between locations, and thus is a minimum dispersal distance. We tested if there was a sex-bias in dispersal distance using a Mann–Whitney U-test. The direction of dispersal was calculated by the degrees of each straight line, considering four sectors—316°–45° (N), 46°–135° (E), 136°–225° (S), 226°–315° (W)—and tested for differences by sex with a Fisher’s Exact test. By reconstructing the pedigree of each stable pack in the area, each genotype was categorized as a “pup/offspring”, or an “alpha/parent”, or as “other”, if it was not related to any pack, and each frequency of status change was calculated. Tests and graphs were performed in R version 4.1.2. [40], using RStudio v2021 [41]. All values reported in the Results are means ± SD.

## 3. Results

### 3.1. Genotyping, Identification of Recapture, and Genetic Variation

From 2001 to 2021 we analyzed 5729 genetic samples. Of these, 142 were identified as being from a species other than wolf: 106 domestic dog (*Canis familiaris*), 33 red fox (*Vulpes vulpes*), and three jackal (*Canis aureus*). Of the remaining 5587 wolf samples, we successfully genotyped 2857 (51.1%) and identified 915 unique individuals. Genotyping success rate varied among sample types ranging from 99.1% for tissues, 49.4% for scat, 44.4% for hair, and 15.4% for saliva swabs. Using the microsatellite data, we calculated the probability of identity (PI) [42] and the probability that siblings are identical (PIsib) [43] to determine the power in the dataset to distinguish individuals. The calculated PI and PIsib were 3.72 × 10^−12^ and 1.20 × 10^−5^, respectively. We calculated standard measures of genetic variation such as observed heterozygosity (0.58), expected heterozygosity (0.61), mean polymorphic information content (PIC) 0.55, and average number of alleles per locus (7.38; SE 0.64). All wolf samples but two in our dataset were identified as being identical to the common Italian haplotype (W14 in Randi et al. 2000). The two remaining wolf sequences match wolf haplotype W3 previously identified from wolves in Croatia and Slovenia [27,44]. We could reconstruct pack pedigrees for the majority of genotypes [6], and we documented 55 individuals that were recaptured in different areas and moved from one pack territory to another, often changing social status. We defined these 55 documented cases as dispersal events, which constitutes 6% of the total individuals identified.

Of the 55 dispersing wolves identified in our study, 27 were identified as pups that then moved to become alphas of new packs. In some of these cases, we were able to explore how the dispersing individual may have influenced genetic diversity of the new pack. For example, one male originating from the Dinaric population in Slovenia as a pup (SLO-M01) became the alpha male of a new pack in Lessinia, an Eastern part of the Alps in Italy. The observed heterozygosity of this pack was much higher than expected (Ho = 0.79 compared to He = 0.56). While the average number of alleles per locus was 2.8, this male brought with him 4 alleles not seen before in our wolf samples from Italy, and those alleles have persisted in wolves from that region through 2021. In the Western Alps, observed heterozygosity increased from 0.59 to 0.64 in the Maira pack after the dispersal of CN-M192 (from Valle Stura Bassa; Ho = 0.68) who became the alpha of the pack. We also observed cases where observed heterozygosity and allelic diversity was similar to the packs the dispersing wolves originated from: TO-M57 (Val Chisone; Ho = 0.60; average number of alleles 2.2) and CN-F86 (Pian Regina; Ho = 0.58; average number of alleles 2.9) became the alphas of a new pack in Valle Ripa (Ho = 0.58; average number of alleles 2.9).

### 3.2. Wolf Dispersal Spatial and Individual Characteristics in the Italian Alps

Of the 55 identified dispersing wolves, 27 were males and 28 were females. The minimum distance of dispersal events varied from 7.7 km to 517.2 km, defined as the straight distance among the two detections. Detailed information on the 7 dispersal events over 100 km in length are given in Table 1. All dispersal events started from the source population in the Western part of the Alps, except the two recent events in 2014 detected in the Eastern Alps, of two males which originated from the Dinaric population in Slovenia. On average, the dispersal distance was 65.8 km (SD 67.7 km) (Figure 1). Females dispersed shorter straight distances than males (Mann–Whitney U test: W = 261, *p* = 0.049, 2-tailed) (Figure 2). Males dispersed, on average, 93.6 km (median 57.8 km, IQR: 23.2–87.8); females dispersed, on average, 47.8 km (median 28.0 km, IQR: 17.7–51.3). The fate of dispersers varied, with the majority of wolves that dispersed from their natal pack occupying alpha positions and reproducing in their new pack (51%, 28 out of 55) (Figure 3). The frequency of wolves dispersing in each directional sector did not differ (χ2 = 5.7, df = 3; *p* = 0.128), even between males and females (Fisher’s Exact Test *p* = 0.127) (Figure 4).

## 4. Discussion

### 4.1. Wolf Dispersal Patterns in the Italian Alps Documented with Non-Invasive Genetic Analysis

*Distance.* We used data from a non-invasive molecular genetic monitoring program of wolves, to show that wolves in Italy occasionally conduct long distance dispersal movements. In fact, we had 7 dispersal events over 100 km in length. This supports the hypothesis that individuals may often attempt to colonize far from their native pack, as documented in other studies in North America [10,11] and northern Europe [12,13]. In our study, however, the majority of successful documented dispersers who formed a new pack dispersed for short distances, an average of 65.8 km (±67.7 km). Moreover, we detected a slight sex difference in straight-line distance length, where females dispersed shorter distances then males. In a recent review, Morales-González et al. (2022) [20] showed that sex differences in dispersal distance only occurred in some populations worldwide, with males showing higher rates of dispersal and longer travel. Our non-invasive genetic monitoring approach allowed us to not only document dispersing events at the population level, but also to have indications on the success of the documented cases, considering the pedigree analysis on the former packs, which indicated that more than half of the wolves successfully bred in a new pack. The majority of these status changes to alphas were females. Two wolves that we detected with our genotyping dispersal study were also monitored with radio collars from the Apennine population [45], and Dinaric population [8] to the alpine one (Figure 1). The documented cumulative line distance of these two dispersal events were 958 km [45] and 1176 km [8], respectively, showing that the minimum straight-line distance length is shorter (239 and 233 km, respectively), and should be taken as a minimum index of the movement. However, the majority of dispersal studies report straight-line distances, also if documented by radio tracking [20], making our results highly comparable.

*Direction.* The directions of dispersal indicate that wolves in the Western Alps are moving in any direction, but primarily along the north–south axis for long distance movements, where the mountain chain is present, slightly towards less density areas present in the north compared to the south, where the recolonization process started [24]. However, we documented more wolves than expected that moved towards higher wolf density areas in the south, or to the east. According to the literature, the dispersal direction might be influenced by individual, environmental, or even social factors. Individual experience can play a role, inducing the dispersing wolves to select habitats for territory establishment similar to their original natal site [46,47]. This might happen especially for short-distance dispersals. Sanz-Pérez et al. (2018) [47] documented that some dispersing wolves selected the highest wolf densities areas for territory establishment, and some studies [9,10] also reported frequent dispersals’ events from colonizing populations to source populations. Hence, this pattern we also observed is not uncommon, especially if we consider the shape of the mountain chain in the Western area, which might induce this pattern. It remains that the majority of studies indicate that dispersal direction is strongly influenced by the risk of interaction with humans [20], which is also showed in the present study by wolves avoiding the highly urbanized planes and dispersing towards territories in forested and mountainous areas with less human population density [16,47,48]. Mountains constitute the majority of the wolf-occupied area in the Western Alps [24], and likely constitute the habitat corridor facilitating similar dispersal routes among individuals, as seen for other species [49].

*Genetic variation*. We were encouraged to see that long distance dispersers were bringing in new alleles and heterozygosity was increasing in those packs formed with a disperser. There have been many studies that have shown that genes from dispersers can “rescue” small populations by reducing genetic load and thus increasing population fitness [50,51]. In fact, genetic rescue has been documented in multiple taxa, including invertebrates, fish, mammals, birds, and reptiles [52]. One of the first documented cases of genetic rescue in mammals was in a re-founded Scandinavian wolf population. There, a single immigrant from Finland was sufficient to bring in enough genetic variation to bolster the growth rate of the population [53]. Interestingly, this same population again benefited from immigration with a second genetic rescue event when two immigrant wolves reached the population and established territories with local females, producing litters for three consecutive years [54].

We encourage the continued genetic monitoring of these wolf packs using one of several methods shown to be helpful in monitoring genetic rescue, or through the use of multiple metrics collected simultaneously [55,56]. Among important metrics to monitor would be population growth or other population fitness measures, yet simply monitoring the number of immigrants coming into the population and having insights into their unique genetic profiles could provide an index of a healthy metapopulation. Fortunately, with the current non-invasive genetic monitoring program, obtaining these metrics is feasible.

### 4.2. Implications for Wildlife Diseases Spreading

Wild canids are involved in the maintenance and spread of major zoonoses of infectious and parasitic etiology, including rabies, echinococcosis/hydatidosis by *Echinococcus granulosus*, alveolar echinococcosis by *Echinococcus multilocularis*, and trichinellosis [57,58]. In the particular case of sylvatic rabies in Europe, the dispersal of young foxes (*Vulpes vulpes*) was shown to be a key determinant of the wavefront advancement speed, in the range of 20 to 60 km/year, with maxima of 100 km/year [59,60]. Interestingly, modeling highlighted that both neighborhood infection and long-distance infection are needed to generate the wave-like dispersal pattern of the disease [61]. Sylvatic rabies, which is exhaustively monitored in Europe (https://www.who-rabies-bulletin.org/, accessed on 13 April 2022), has not been reported in Italy since 2011 [62], and no cases were diagnosed in wolves since they returned in the northern part of the country after extirpation. The longest straight-line dispersal distance documented in this study (517.2 km) is still within the distance between the eastern portion of Northern Italy and the closest sylvatic rabies foci in the Balkans [63]. Nevertheless, a hypothetical reintroduction of rabies by a wolf dispersing during the latent phase of the disease seems unlikely for several reasons, including the infrequent wolf rabies caseload in Europe [64] and the longer wolf dispersal duration compared with the length of the latent phase of the disease in canids, in the order of 4 to 5 weeks [20,59].

Much less is known, at the continental scale, on the role that wolves play in the epidemiology of major parasitic zoonoses, partly because wolf return or recovery was a recent event in several countries [4]. Wolves are regarded as a potentially relevant host considering the introduction of the fox tapeworm, *E. multilocularis*, a deadly though rare pathogen in humans, into areas that are deemed free of the parasite [65]. Several surveys have documented the ongoing geographical spread of *E. multilocularis*, peripheral to the historically endemic areas in Central Europe [66]. In Northern Italy, a single endemic area has been recognized since the early 2000 in the Eastern Alps, at the border with Austria [67]. When it comes to the study area, no cases have been recorded in humans nor was the adult tapeworm found in the digestive tract of 42 necropsied wolves [68], but a recent survey documented the unexpected occurrence of *E. multilocularis* DNA in the feces of five wolves at the southern edge of the Western Alps, 130 km south of the closest endemic area in the Hautes-Alpes, France [69]. This distance well fits the medium-range dispersal distance of wolves in this study. The life expectancy of adult *E. multilocularis*, in its definitive hosts, is in the order of a few months in foxes, though longer in dogs [70], and is also compatible with wolf dispersal duration in European landscapes [20]. Accordingly, dedicated studies on the potential relationships between wolf return and the spreading of *E. multilocularis* at the edges of its distribution area in Southern Europe seem all the more advisable. A similar concern applies to the possible role of dispersing wolves in spreading the dog tapeworm, *E. granulosus*, from the endemic peninsular Italy and southeastern France to the hypoedemic northern Italy, with increased risk for farmers and the rural communities in general [71] (https://www.anses.fr/fr/content/bulletin-echinote, accessed on 13 April 2022). Data from the Northern Apennines, the source of dispersing wolves that recolonized Northwestern Italy, showed that 5.5 to 26.4% of examined wolf scats tested positive for *E. granulosus* DNA [72,73,74], highlighting sustained wolf exposure to the “domestic” (livestock-dog) transmission cycle. Of note, a “sylvatic” cycle involving wolf and the main prey as principal definitive and intermediate hosts of *E. granulosus*, respectively, has never been documented in Southern Europe [75,76].

Wolves have been hypothesized as a possible contributor to the spread of African Swine Fever (ASF), a severe transmissible disease of domestic and wild swine, with a tremendous socio-economic impact due to eradication measures provided for by international legislation. The virus agent of ASF is able to persist for a long time in meat, blood, and the carcass environment [77]. In Europe, wild boar is considered a major and long-term reservoir, with self-sustaining infectious cycles that may result in spill-over episodes in pig farms. Since 2007, ASF has been spreading amongst wild boars in several eastern European countries, in a limited portion of Belgium, and recently in Germany [78]. The last reported outbreak (index case in January 2022) is developing in Italy at the south-eastern edge of our study area (https://www.izsplv.it/it/notizie/308-peste-suina-africana/, accessed on 13 April 2022). There is a popular debate amongst many people and in the media on the role those dispersing wolves play in the long-range spreading of ASF virus and the assumed greater difficulties in eradicating wild-boar ASF where wolves are roaming. These undocumented speculations may negatively affect the public attitude towards wolves in rural communities suffering restrictions in line with ASF eradication policies [79]. Data in this study suggest that average wolf dispersal distance (65.8 ± 67.7 km) is remarkably longer than the expected advancement of wild-boar ASF wavefront, reportedly ranging between 8 and 25 km/year [80]. This gives strong support to the opinion that determinants other than wolf dispersal, related to inappropriate or illegal human behavior, are primarily involved in ASFV long-range dispersal. In addition, ASFV DNA was not traceable in the feces of GPS-collared wolves scavenging on ASF infected wild boars [79]. Finally, the long-distance transportation of viable ASFV by a wolf dirty with blood or other fluids after scavenging an infected wild boar carcass seems unlikely due to grooming. This remote risk is clearly outweighed by the efficient removal of infected wild boar carcasses and remains.

Another interesting output of this study is that a non-negligible number of individuals have dispersed across intensely anthropized habitats, apparently without avoiding them. Not surprisingly, human density has been identified as a major driver of pathogen exposure in wolves in North America and Europe. In fact, human density may be used as a proxy for density of unvaccinated dogs, a primary reservoir for many transmissible diseases and parasites with the potential to spill over into wolves [81]. Amongst them, distemper by CDV is the most feared viral agent from a wolf conservation perspective [82]. Since their restoration, wolves in Northern Italy were only marginally involved in the multiple and severe CDV outbreaks that affected other wild carnivores (*V. vulpes*, *Meles meles*, *Martes foina*) on both sides of the Alps. The agents of these outbreaks were identified as “European wildlife-like” CDV strains, deemed typical of wildlife [83]. However, a deadly though localized distemper outbreak caused by an “Artic-like” strain of CDV, typically found in dogs, has been recorded in wolves in Central Italy [84]. The occurrence of wolves in intensively anthropized habitats in Northern Italy is the precondition for enhanced encounters with unvaccinated free-roaming dogs, whose minimum number (corresponding to captured dogs admitted to public kennels) has been estimated in 2020 on the order of 32,000 individuals (https://www.salute.gov.it/portale/caniGatti/, accessed on 13 April 2022). Other potential health concerns are those dog-derived macroparasites that may impact on the fitness of individual wolves. Two of them, the vector-transmitted heartworm (*Dirofilaria immitis*) and eyeworm (*Thelazia callipaeda*), have been recorded in wolves originating from anthropized low altitude locations within the study area [85,86].

## 5. Conclusions

This multidisciplinary work highlights the ecological importance of long-term monitoring, particularly for wide-ranging carnivores. Our ability to identify wolf dispersers is due to an extensive temporal and spatial dataset. Knowledge of dispersal features by means of long-term, not invasive, genetic tools is key information for monitoring genetic rescue and other important metrics of population fitness that could provide indexes of a healthy metapopulation, so our approach is particularly suitable in this context and can be used elsewhere. Non-invasive genetic tools demonstrated to be a very successful and largely applicable technique, which allowed us to follow not only the spatial demographics of dispersers, but also to document the maintenance of adequate heterozygosity levels through dispersal and subsequent mating, confirming this to be a technique which has widespread functional applications to a variety of elusive carnivore species. In this context, the non-invasive genetic tools appear to be a more comprehensive technique which allow the investigation of, at the same time, spatial, demographic, and genetic patterns in a large-scale distributed population, compared to the more traditional telemetry tools. The approach is also particularly useful for providing information for the modelling of major transmissible diseases of wildlife, impacting on human and animal health, livestock economy, and biodiversity conservation. On turn, adequately informed disease modelling may drive the decisions of competent agencies on: (i) the optimal use of budget and human resources for surveillance; (ii) the implementation of preventive and control measures, if desirable, that are ethically admissible and economically sustainable. In contrast with the popular image of wolves as long-range spreaders of much-feared pathogens, examples discussed suggest that in Northern Italy, where natural spaces alternate with densely populated areas, surveillance of major diseases and parasites whose spreading could be favored by dispersing wolves should prioritize: (i) relatively shallow buffer areas in proximity of known disease foci (e.g., in the case of *E. multilocularis* and *E.granulosus*); (ii) wolves settled at the edge of their distribution areas, usually in an anthropized zone at lower altitude where interactions with roaming unvaccinated dogs and dog-contaminated environments may occur at higher frequency than in remote zones, eventually resulting in the spill-over of carnivore-specific pathogens (e.g., in the case of CDV).

## Figures and Tables

**Figure 1 animals-12-01260-f001:**
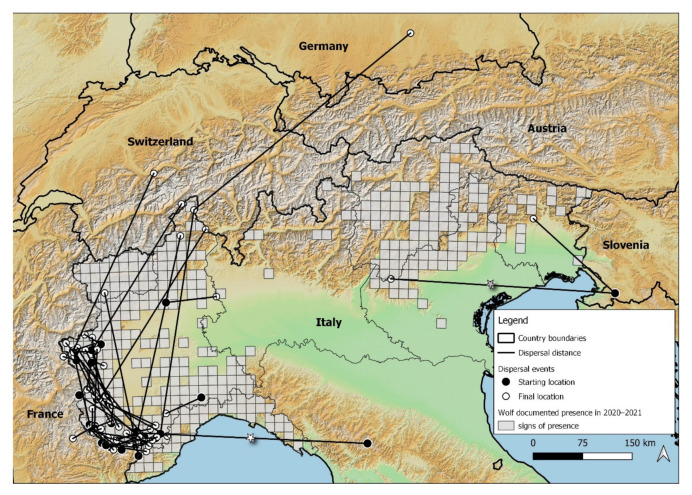
Distribution of dispersal events over the Alps documented by non-invasive genetic analysis over 20 years (from 2001 to 2021). The stars indicate dispersal events also documented by means of GPS collars in two published studies, cited for comparison to the genetic tools used in this study; the white star by [45], the grey star by [8].

**Figure 2 animals-12-01260-f002:**
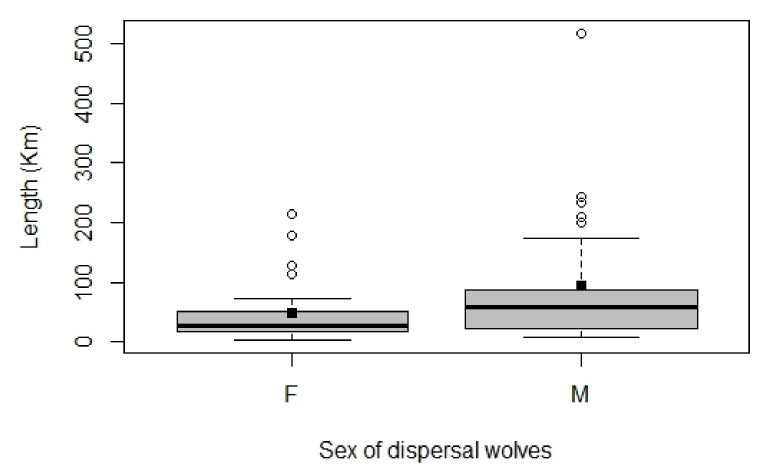
Boxplot of dispersal distances by sex. The highlighted bar represents the median of the sample, the width of the box is the interquartile range (IQR). Outliers are indicated with circles. Black squares indicate the mean.

**Figure 3 animals-12-01260-f003:**
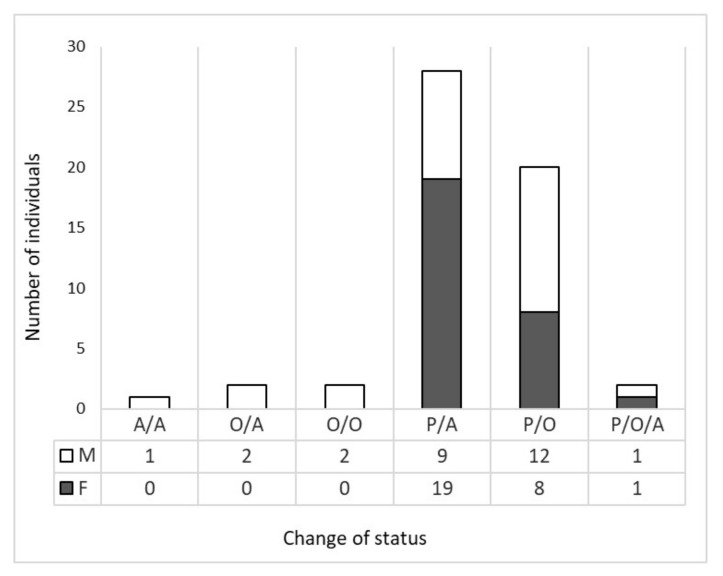
Frequency of changes in social status after dispersal events. By reconstructing the pedigree of each stable pack in the area, each genotype was categorized as a “pup/offspring” (P), or an “alpha/parent” (A), or as “other” (O), if it was not related to any pack, and each frequency of status change was calculated by sex.

**Figure 4 animals-12-01260-f004:**
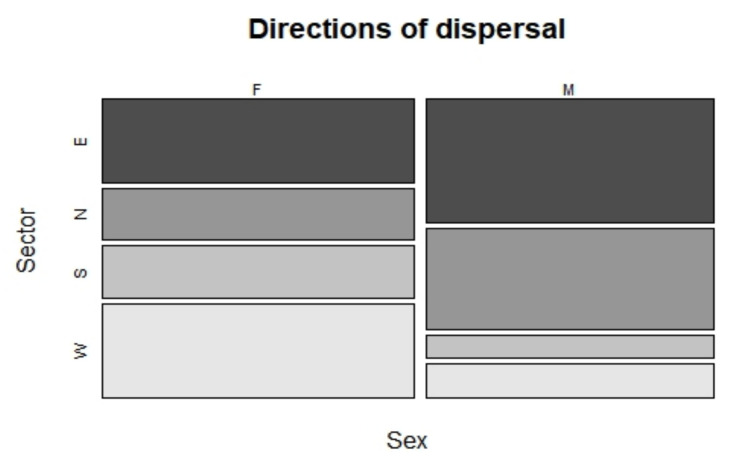
Mosaic plot of the frequency of wolves dispersing in each directional sector by sexes. Sectors are defined as follows: 316°–45° (N), 46°–135° (E), 136°–225° (S), 226°–315° (W).

**Table 1 animals-12-01260-t001:** Detailed information on the 7 dispersal events over 100 km in length. The area of provenience and arrival is indicated (the 2 letters indicate the Italian province), as is the status (P: pup/offspring, A: alpha/parent, O: other), and if the animal has been recovered dead.

ID Genotype	Sex	Length of Dispersal (km)	Area of Dispersal		Change of Status		Recovered Dead
			From	To	From	To	
CN-46	M	174.4	CN (IT)	AO (IT)	P	O	No
TO-46	F	177.7	TO (IT)	Swiss	P	O	Yes
TO-41	M	199.5	TO (IT)	Swiss	P	O	Yes
CN-123	M	208.6	CN (IT)	VB (IT)	P	O	No
CN-31	F	214.1	CN (IT)	VB (IT)	P	O	No
SLO-01—Slavc	M	233.0	Slovenia	VR (IT)	P	A	No
CN-95-Ligabue	M	239.0	PR (IT)	CN (IT)	P	O	Yes
CN-100	M	517.2	CN (IT)	Germany	P	O	Yes

## Data Availability

Not applicable.

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
