# Peer review of "Wolf Dispersal Patterns in the Italian Alps and Implications for Wildlife Diseases Spreading"

_animals, 2022, doi:10.3390/ani12101260_

Round 1

Reviewer 1 Report

Line 21, suggest using (range: 7.7 to 517.2 km)

note: my initial immediate query is how dispersal distance can be calculated using only genetic sampling, instead of active animal-mounted collars and tracking? No doubt this will be explained soon! 

A second initial question = how do you know this is dispersal vs. a home range? 7,7km for a wolf does not seem like a particularly large distance to travel, so why would it be termed dispersal?

Line 58: I think this would be perfectly displayed with a map-based image, showing areas colonised if such data were available, which they appear to be?

Line 65: It would be good to mention here if dispersal were sex biased or not. Is there male or female philopatry or none at all? 

Line 85: I think the percentages are unnecessary the way this is written. It would be better to say where 98% were feces, 0.38% hair and the rest tissue from dead animals. 

Line 85: genetically analysed replace with “genotyped”

Line 87: How many hairs were included per extraction? Were any extractions done in duplicate? Elutions in 100uL is a bit brave, because usually that would make these pretty dilute, so I’m curious what your DNA yields were post extraction?

Line 99: What does this mean? What quality and quantity? You also mention DNA template meaning this is the what goes into the PCR, so this sentence would need to come before the PCR details. Or perhaos you meant that the DNA amplicon produced after the PCR is what was assessed on a gel? 

Line 94: please give the expected size of the amplicon.

Line 101: Sanger sequenced in both directions or only one? 

Line 108: Please elaborate and don’t just cite 33. Tested for sex how? How many times were “negatives” repeated to confirm them?

Line 128: Oen thing not mentioned here is how territories and packs are defined. What level of monitoring was needed for this? How certain are we of territories for all the packs across this 20 year period? What kind of monitoring effort was that?

Line 130: I’m having some trouble understanding how this is not biased. If there are two home ranges for a pack, and the animals move from one to another, shouldn't the dispersal distance be recorded as the centroid to centroid distance? By simply taking the straight line distance between where the animal’s feces was deposited you can be artificially inflating or deflating this dispersal distance just by chance. Had you picked up a fecal sample on a different day in a different location within the pack;s home range, it’s dispersal distsnce would be totally different. 

Line 140: So only 50% of the samples yielded DNA? This is pretty low, and were there differences by tissue type as well in DNA extraction success? It would be useful to know which tissues yielde the best outputs as well.

Line 145: for thid average please provide an SD

Line 161: so very interesting!

Line 171: This is pretty telling…if SD is larger than the mean itself, then you have a hugely varying number. I think this reflects the inherent error in documenting dispersal distances the way you have. 

Line 174: grammar here is a bit strained. Reproducer, not reproductor. And you might say” The fate of dispersers vaired, with the majority of wolves that dispersed from their natal packs occupying alpha positions and reporducing in their new packs”. Althoug, 51% is hardly the majority…its basically just half. 

Line 176: “even between males and females"

Figure 2: Length is spelled incorrectly in the y axis. 

Figure 1: I think it would be very interesting to see the zoomeed in area on the bottom left, with the home ranges of a pack indicated that show how tightly packed they are and why some wolves only dispersed 7.7km to get to a new pack. Is this distance smaller than the average home range?

Line 244: from where?

Line 272: so this is not from your study? The dispersal duration?

Line 300: My biggest concern here is that your analysis, although it could have, says nothing about diseases. These could have been screened in the same fecal or tissue samples and data prodiced to bolster your argument that it is important to study wolf disease spread. It would have been so much more effective to see if any of the diseases could have been tested for in the samples you have because otherwise, this is effectively just a review of existing literature on wolf diseases and not necessarily something new? My advice would be to not center disease as much, you do not need it to justify the study, but to protray it as an important implication for this work. 

Reviewer 2 Report

The manuscript describes population genetics analyses of wolves in the Italian Alps using STR markers focusing on the dispersal of the animals. Dispersal events found were described and discussed with the potential implications for maintaining genetic diversity and for wildlife diseases spreading. The sampling is impressive, with a large number of mainly non-invasive samples collected between 2001 and 2021. The manuscript is deliberate and mainly well written. I have only some minor suggestions and questions.

Based on the text of the Materials and Methods section, sampling occurred “across the Italian Alps”, but in Figure 1. there are some samples depicted outside the Alpine regions, and even outside Italy. Were samples collected in other countries also included? This should be clearly described.

Please be consistent by naming manufacturers and other providers.

You should be also consistent with mentioning the version number and referencing the source of the software used. CERVUS, QGIS and R lack a reference. Although the homepage of CERVUS is mentioned the original publication is not. Two versions of GenAlEx were referenced, the newer one would suffice.

It is mentioned in the Results section, that the directionality of dispersal events was analysed using Fisher’s Exact Test, but this is completely omitted from the Methods. Please add a little description of this in the Methods, and the results of the tests should be clearly presented. Based on Figure 1., the majority of the dispersal occurred in north-western Italy, but some also in north-eastern Italy. Were the western and eastern events combined for the analyses? As different constraints are present in the two regions, it would be interesting to investigate them also separately.

It is hard to see exact distance values based on Figure 2. You could indicate average, median dispersal and IQR for the sexes separately in the text or the figure, or make the y axis larger. The bar plots in Figure 3. could also be supplemented with the number of individuals.

In line 197 “7 dispersal events over 100 km in length” were mentioned. I would be interested in where these occurred, i.e. from where to where and what distance did the animals move. Maybe a table could be added with this information.

My final question is, are the individual wolf genotypes available somewhere, and if not do you plan to make them available?

Reviewer 3 Report

A highly relevant manuscript, documenting several interesting findings tied to dispersal as detected through DNA sampling, rather than the more typical use of telemetry collars.  Parameters discussed include the potential disease spread, demographics of dispersers, the value of long-term research (2o0 yrs in this study), maintenance of adequate heterozygosity levels through dispersal and subsequent matings. This technique has widespread functional applications to a variety of elusive carnivore species.  It may strengthen the manuscript to have a longer discussion section citing what has and has not been discovered using comparable DNA methodology vs more traditional telemetry in wolves and even other wide-ranging carnivore species, and the efficacy of DNA vs telemetry in Europe and North America.  This would be helpful to agencies and managers for monitoring and management of their wolf populations.

I detected some minor language use problems and typos that can be corrected by your editing staff.

Overall, an interesting paper, clearly written, and a sound contribution to the literature.
